# Text-to-CAD Generation Through Infusing Visual Feedback in Large Language Models

**Ruiyu Wang** [1][†]   **Yu Yuan** [2][†]   **Shizhao Sun** [3]   **Jiang Bian** [3]

## Abstract

Creating Computer-Aided Design (CAD) models requires considerable expertise and effort. Text-to-CAD, which converts textual descriptions into CAD parametric sequences, is crucial in streamlining this process. Recent studies have utilized ground-truth parametric sequences, known as sequential signals, as supervision to achieve this goal. However, CAD models are inherently multimodal, comprising parametric sequences and corresponding rendered visual objects. Besides, the rendering process from parametric sequences to visual objects is many-to-one. Therefore, both sequential and visual signals are critical for effective training. In this work, we introduce CADFusion, a framework that uses Large Language Models (LLMs) as the backbone and alternates between two training stages: the sequential learning (SL) stage and the visual feedback (VF) stage. In the SL stage, we train LLMs using ground-truth parametric sequences, enabling the generation of logically coherent parametric sequences. In the VF stage, we reward parametric sequences that render into visually preferred objects and penalize those that do not, allowing LLMs to learn how rendered visual objects are perceived and evaluated. These two stages alternate throughout the training, ensuring balanced learning and preserving benefits of both signals. Experiments demonstrate that CAD-Fusion improves performance, both qualitatively and quantitatively. Code is available at `https://github.com/microsoft/CADFusion`.

---

[†] Work done during the authors' internship at Microsoft Research Asia. Open-source research project starts at March 2024. Ruiyu Wang: <rwang@cs.toronto.edu>, Yu Yuan: <yy-happier@mail.ustc.edu.cn>, Jiang Bian: <jiabia@microsoft.com>. [1]University of Toronto. [2]University of Science and Technology of China. [3]Microsoft Research Asia. Correspondence to: Shizhao Sun <shizsu@microsoft.com>.

*Proceedings of the 42nd International Conference on Machine Learning*, Vancouver, Canada. PMLR 267, 2025. Copyright 2025 by the author(s).

## 1. Introduction

Computer-Aided Design (CAD) is indispensable for 3D creation across industrial sectors (Deng et al., 2023). It represents 3D models through a sequence of operations known as a *parametric sequence*, which combines lines, arcs, and circles to create 2D sketches and then extrude them to form 3D models. CAD models are inherently multimodal, as they are constructed using parametric sequences for precise editing and manufacturing, while also being rendered as visual objects for practical use, referred to as *multimodal characteristic* (Figure 1(b)(c)). Moreover, the process of rendering parametric sequences into visual objects exhibits a many-to-one mapping, where different parametric sequences can result in identical visual objects, referred to as *many-to-one rendering characteristic* (Figure 1(d)).

Creating CAD models demands considerable expertise and numerous iterations, making it complex and time-consuming. *Text-to-CAD* (Figure 1(a)(b)), which refers to the automatic generation of parametric sequences from textual descriptions, is critical for streamlining this creation process. It allows designers and engineers to quickly prototype and iterate designs by describing their intent in natural language, reducing the time spent on manually creating CAD models from scratch. Additionally, it makes the creation process more accessible to individuals without extensive training, enabling wider participation.

While important, Text-to-CAD has received limited attention. Most studies do not utilize text to control CAD generation. Instead, they explore generating CAD designs from random noise (Wu et al., 2021; Xu et al., 2022), by randomly altering components of existing CAD designs (Xu et al., 2022; 2023; Zhang et al., 2025), or from point cloud (Khan et al., 2024a). A few studies make preliminary attempts at Text-to-CAD (Khan et al., 2024b; Li et al., 2024b). They train Transformer-based framework with ground-truth parametric sequences as supervision, termed *sequential signal*.

However, due to multimodal and many-to-one rendering characteristic of CAD models (Figure 1(b)(c)(d)), both the *sequential signal* and *visual signal* are crucial for training a Text-to-CAD model. The sequential signal, derived from ground-truth parametric sequences, provides critical infor-

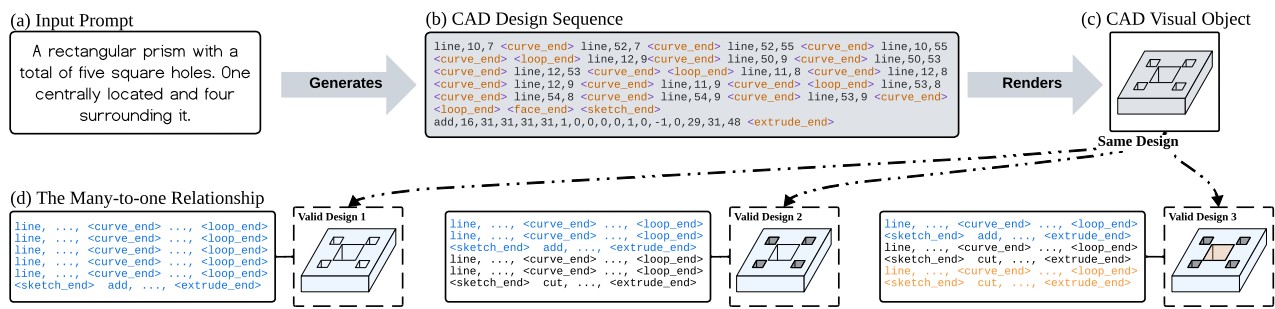

*Figure 1.* **(a) and (b):** Illustration of Text-to-CAD, which converts a textual description into CAD parametric sequences. **(b) and (c):** Illustration of multimodal characteristics. CAD models are created using parametric sequences and rendered as visual objects for practical use. **(d):** Illustration of many-to-one rendering characteristics. Different parametric sequences can produce identical visual objects.

mation about sequence structure and parametric operations. Without it, learning to generate logically coherent parametric sequences becomes challenging, as there is no direct supervision for sequence structure and parametric operations. The visual signal, obtained from rendered visual objects, indicates how CAD models are perceived and evaluated in practical applications. Without it, learning efficiency is compromised, as the goal of Text-to-CAD is for the rendered visual objects of the generated parametric sequences to match ground-truth visual objects. First, sequential signal learning typically depends on auto-regressive generation, which emphasizes the local continuity between tokens but may not fully capture the global appearance of the CAD model. Second, given the many-to-one rendering characteristic, multiple parametric sequences can produce the same visual object. Training solely on parametric sequences may cause the model to give more emphasis to those present in the training set, overlooking other valid ones that could achieve the same visual outcome.

To this end, we propose CADFusion, a framework that combines sequential and visual signals to train a Text-to-CAD model. It uses Large Language Models (LLMs) as its backbone and alternates between two stages: the *sequential learning stage* and the *visual feedback stage*. In the sequential learning stage, LLMs are fine-tuned using ground-truth parametric sequences. Unlike prior works (Khan et al., 2024b; Li et al., 2024b) that train Transformer-based models from scratch, we take advantage of pre-trained LLMs, which leverages their inherent natural language understanding and foundational knowledge of CAD design (Makatura et al., 2023) acquired during the extensive pre-training. In the visual feedback stage, feedback derived from rendered visual objects is integrated into the LLMs. This stage addresses two critical challenges. First, the rendering process that converts parametric sequences into visual objects is non-differentiable, making backpropagation through this pathway infeasible. To overcome this, we frame the problem as preference learning task and adopt direct preference opti-

mization (DPO) (Rafailov et al., 2024). Specifically, preferences are assigned to the rendered visual objects, and the LLMs are optimized to increase the likelihood of parametric sequences that produce preferred visual objects while decreasing the likelihood of those that yield less preferred ones. This approach enables effective training of LLMs, even with a non-differentiable rendering pathway. Second, collecting reliable preference data is costly and labor-intensive. To address this, we introduce an automated pipeline that utilizes large vision-language models (LVMs) to efficiently score the rendered visual objects. Finally, to ensure balanced learning and retain the contributions of both signals, we alternate between the sequential learning stage and the visual feedback stage throughout training.

We summarize our main contributions as follows:

- We propose to leverage both the sequential signal and visual signal to train a Text-to-CAD model.
- For the sequential signal, we use LLMs as the backbone and fine-tune it on ground-truth parametric sequences. For the visual signal, we adopt direct preference optimization to bypass non-differentiable rendering and introduce a LVM-based scoring pipeline for efficient preference data collection. To balance both signals, we alternate between the sequential learning and the visual feedback stage.
- We contribute two datasets for Text-to-CAD: one with the sequential signal and another with the visual signal.
- We present qualitative and quantitative experiments to showcase CADFusion's superior ability.

## 2. Related Works

**CAD Generation.** CAD generation takes user requirements as input and generates CAD models as output.

On the input side, user requirements can be expressed in diverse ways. Wu et al. (2021) uses random noise as input to generate CAD models randomly. Zhang et al. (2025),

Xu et al. (2022) and Xu et al. (2023) modify specific parts of the existing CAD models to generate new ones. Khan et al. (2024a) and Ma et al. (2024) take point cloud as input to produce corresponding CAD models, while Seff et al. (2022) uses hand sketches. In contrast, our work focuses on textual descriptions as input, leverages LLM as backbone on stringified representations, and does not adapt codebook encoders such as VQ-VAE to generate initial representations. Recent studies (Khan et al., 2024b; Li et al., 2024b) explore text-based input for CAD generation. Khan et al. (2024b) proposes a data annotation pipeline for synthesizing training data and a transformer-based autoregressive network. Li et al. (2024b) designs an encoder-decoder framework with a cascading contrastive strategy and CT-Mix to align text with parametric sequences. Unlike these studies, which rely solely on sequential signals, our work combines sequential and visual signals for improved performance.

On the output side, CAD models can be represented in various formats, including Constructive Solid Geometry (CSG), Boundary Representation (B-Rep) and Sketch-and-Extrude (SE). CSG constructs 3D models by combining basic primitives such as cubes, cylinders, and spheres, through Boolean operations and subtractions (Du et al., 2018; Kania et al., 2020; Yu et al., 2022; 2024). B-Rep represents 3D models using geometric elements such as vertices, edges, and faces (Jayaraman et al., 2023; Wang et al., 2022; Xu et al., 2024). SE begins with 2D sketches composed of lines, arcs, and circles, which are then extruded to form 3D models (Willis et al., 2021; Wu et al., 2021). In this work, we adopt SE as it preserves the design history of CAD models, making them more intuitive to edit.

**Large Language Models (LLMs).** LLMs have recently achieved remarkable success (Touvron et al., 2023; Brown et al., 2020; OpenAI, 2024; Bubeck et al., 2023; Zhao et al., 2023). Supervised fine-tuning (SFT) is widely used to improve performance, while reinforcement learning (RL) is often employed to align LLM output with human preference (Brown et al., 2020; Hong et al., 2024; Kaufmann et al., 2024). Our work leverages SFT and RL[1] but introduces two key differences. First, we utilize SFT and RL to learn from different signals (i.e., sequential and visual signals) whereas existing work focuses on a single signal (i.e., sequential signals). Second, we alternate between SFT and RL stages to preserve contributions from both signals, a strategy not commonly employed in prior work.

**Reinforcement Learning with Human Feedback (RLHF).** RLHF has been widely applied to align model output with human preference across various domains, including LLMs (Brown et al., 2020; Radford et al., 2021; Meta, 2024),

---

[1]We adopt DPO (Rafailov et al., 2024) in practice and refer to it as RL here for simplicity, as it implicitly optimizes the same objective as traditional RLHF despite not being a typical RL.

text-to-image models (Liang et al., 2024) and text-to-video models (Wu et al., 2024). As human annotation in RLHF is costly and not easily scalable, reinforcement learning on AI feedback (RLAIF) (Liu et al., 2023; Zhang et al., 2024; Lee et al., 2024), which leverages machine learning models to annotate data, has been proposed as a more affordable alternative to RLHF. Since RLHF/RLAIF pipeline are complex, direct preference optimization (DPO) (Rafailov et al., 2024), which directly optimize a model to adhere to human preferences, has been proposed to avoid explicit reward modeling or reinforcement learning. In this work, we adopt DPO to address the challenge of non-differentiable rendering when learning from visual signals, as it offers a simpler yet effective solution compared to RLHF. Besides, inspired by RLAIF, we propose an automatic scoring pipeline for CAD models using LVMs. The generated scores are used to construct preference data, enabling efficient learning without reliance on costly human annotations.

## 3. Method

### 3.1. Approach Overview

Let a textual description be denoted as $x$, a CAD parametric sequence as $y$, and a rendered visual object as $o$. The rendering process from a parametric sequence $y$ to a visual object $o$ is represented as $r(\cdot)$, such that $o = r(y)$. Text-to-CAD involves learning a function $f(\cdot)$ that transforms the textual description $x$ into the CAD parametric sequence $y$. i.e., $y = f(x)$. The goal is for the rendered visual object $o$ of the generated parametric sequence $y$, i.e., $o = r(y) = r(f(x))$, to match the user's desired visual object (Figure 1).

CADFusion introduces a framework that combines sequential and visual signal for training a Text-to-CAD model (Figure 2). It leverages Large Language Models (LLMs) as the backbone and alternates between two stages: the *sequential learning (SL) stage* and the *visual feedback (VF) stage*. We denote the model after the $i$-th round of sequential learning as $f_{SL}^i(\cdot)$ and after the $i$-th round of visual feedback as $f_{VF}^i(\cdot)$. In the sequential learning stage, CADFusion trains LLMs to learn sequence structures and parametric operations from ground-truth parametric sequences, guiding LLMs to generate logically coherent parametric sequences (Section 3.2). In the visual feedback stage, CADFusion trains LLMs to understand how the rendered visual object will be perceived and evaluated. By rewarding parametric sequences that render into visually preferred objects and penalizing those that do not, this stage encourages LLMs to generate parametric sequences capable of producing the desired visual object (Section 3.3). These two stages are alternated throughout training, ensuring balanced learning and preserving contributions of both signals (Section 3.4).

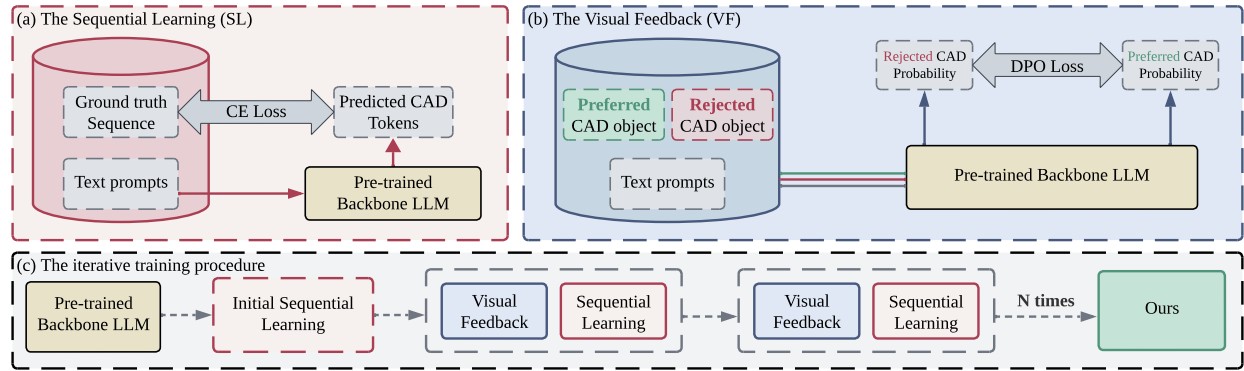

*Figure 2.* Overview of CADFusion. **(a):** The sequential learning stage trains LLMs using ground-truth CAD parametric sequences. **(b):** The visual feedback stage rewards CAD parametric sequences that render into preferred visual objects and penalizes those that do not. **(c):** The two stages are alternated to preserve contributions of both signals.

### 3.2. Sequential Learning Stage

Text-to-CAD requires a model capable of understanding textual descriptions and generating CAD parametric sequences that adhere to valid sequence formats and employ meaningful parametric operations. We adopt the following strategies to efficiently achieve these capabilities.

*1) Model architecture.* We use LLMs as the backbone, leveraging their strong natural language understanding and basic CAD design knowledge (Makatura et al., 2023).

*2) CAD Parametric Sequence Format.* We adopt the format proposed by Zhang et al. (2025) (Figure 1(b)), which represents CAD parametric sequences as text tokens rather than binary representations or numerical attributes (Xu et al., 2022; 2023; Wu et al., 2021). This text-based format simplifies processing and interpretation by LLMs.

*3) Dataset.* Existing CAD datasets (Wu et al., 2021) include CAD parametric sequences but lack paired textual descriptions. To address this, we construct a dataset $\mathcal{D}_{\text{SL}} = \{(x, y)\}_1^M$ ('SL' for sequential learning) containing paired text $x$ and CAD parametric sequences $y$. We first prompt a LVM to generate draft captions for rendered CAD model images and then refine these drafts through human annotation to ensure accuracy and conciseness.

*4) Training.* We fine-tune the pre-trained LLMs by minimizing the discrepancy between the generated parametric sequence $\hat{y} = f_{\text{SL}}^i(x)$ and the ground-truth parametric sequence $y$ using cross entropy loss, denoted as $\mathcal{L}_{\text{SL}}$:

$$\mathcal{L}_{\text{SL}} = -\mathbb{E}_{(x,y) \sim \mathcal{D}_{SL}} \left[ \frac{1}{T} \sum_{t=1}^{T} \log p(\hat{y} = y_t | x) \right], \quad (1)$$

where $T$ is the sequence length and $p(\cdot)$ is the predicted probability of the $t$-th token by the model $f_{\text{SL}}^i(x)$.

While existing studies (Khan et al., 2024b; Li et al., 2024b) also consider sequential signals, CADFusion introduces three distinctions: 1) it uses an LLM backbone to leverage pre-trained knowledge, unlike prior work that trains Transformers from scratch; 2) it represents CAD sequences as text tokens, processed with the LLM's tokenizer, whereas others use custom tokenizers; 3) its training data undergoes human annotation, while prior work relies solely on synthesized data. These enhancements enable it to outperform existing approaches, even without the visual feedback stage.

### 3.3. Visual Feedback Stage

The goal of Text-to-CAD is to ensure the rendered visual object from the generated parametric sequence matches the desired visual object. Relying solely on sequential signals compromises training efficiency (see Section 1). To address this, we incorporate visual feedback into the model already trained on sequential signals (i.e., $f_{\text{SL}}^i(x)$).

**Learning Visual Feedback through DPO.** A straightforward way to incorporate visual feedback is through supervised learning, which minimizes the loss between the rendered visual object from the generated parametric sequence $\hat{o} = r(f(x))$, and the ground-truth visual object $o$. However, since the rendering process $r(\cdot)$ is non-differentiable, this loss cannot be backpropagated to the model $f(\cdot)$. To address this, we reformulate the task as a reward maximization problem, where visual feedback serves as the reward, enabling optimization without requiring a differentiable rendering process. Since conventional RL is computationally expensive, we adopt direct preference optimization (DPO) (Rafailov et al., 2024), a simpler and more efficient approach that implicitly performs reward maximization.

Specifically, we construct a preference dataset $\mathcal{D}_{\text{VF}} = \{(x, o_w, o_l)\}_1^N$ where $o_w$ and $o_l$ are rendered from the para-

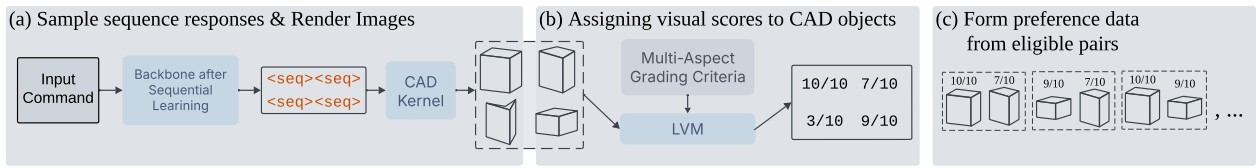

*Figure 3.* Illustration of preference data construction. **(a):** Sample CAD parametric sequences and render them into visual objects. **(b):** Score the visual objects using LVMs with multi-aspect grading criteria. **(c):** Construct preference data based on LVM-generated scores.

metric sequences $y_w$ and $y_l$, representing preferred and less preferred visual objects, respectively. We then optimize the model to increase the likelihood of parametric sequences that produce preferred visual objects ($y_w$), while decreasing the likelihood of those that yield less preferred ones ($y_l$):

$$\mathcal{L}_{\text{VF}} = -\mathbb{E}_{(x,y_w,y_l)\sim\mathcal{D}_{\text{VF}}} \quad (2)$$
$$\left[ \log\sigma(\beta\log\frac{p(\hat{y}=y_w|x)}{p_{\text{ref}}(\hat{y}=y_w|x)} - \beta\log\frac{p(\hat{y}=y_l|x)}{p_{\text{ref}}(\hat{y}=y_l|x)}) \right],$$

where $p(\cdot)$ is the predicted probability of a parametric sequence under the current model ($f_{\text{VF}}^i(x)$), $p_{\text{ref}}(\cdot)$ the probability under the reference model from the last round of sequential learning ($f_{\text{SL}}^i(x)$) and $\beta$ is scaling factor.

**Constructing Preference Data with LVM Scoring.** Collecting preference data is both costly and labor-intensive. The iterative use of the visual feedback stage in our framework (Section 3.4) further highlights the need for a quick and efficient approach for obtaining preference data. To address this, we propose leveraging the strong visual understanding capabilities of LVMs to score visual objects and construct preference data. Figure 3 outlines the pipeline. First, the textual description $x$ is input into the finetuned model after sequential learning ($f_{\text{SL}}^i$) to generate multiple parametric sequences, which are then rendered into visual objects (e.g., CAD images in our implementation). Next, the rendered CAD images, along with an instruction detailing the evaluation criteria, are input into an LVM to obtain scores. Finally, the CAD image with the higher score is regarded as the preferred one (i.e., $o_w$), while the one with the lower score is deemed as the less preferred one (i.e., $o_l$).

Specifically, inspired by recent work (Liang et al., 2024) on evaluating text-to-image generation across rich aspects, we incorporate multiple evaluation criteria into the LVM instruction. As shown in Figure 4, these criteria assess both the appearance of CAD designs and their alignment with textual descriptions: 1) *shape quality* evaluates the regularity, naturalness, and realism of the design; 2) *shape quantity* checks whether the number of components matches the description; and 3) *distribution* ensures components are arranged naturally, avoiding collisions or excessive spacing.

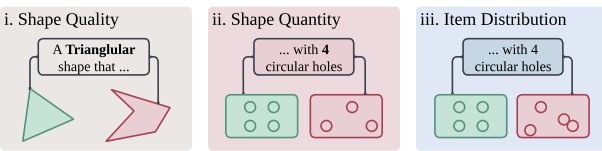

*Figure 4.* An illustrative example of the multi-aspect evaluation criteria used in LVM scoring. Note that the illustrations are simplified to conceptually represent each criterion.

### 3.4. Alternate Training

Each stage of the training process — sequential learning and visual feedback — has a specialized focus. Excessive training in one stage can lead to the degradation of skills acquired in the other. For example, we empirically observe that extended training with visual feedback can impair the model's ability to generate well-formatted parametric sequences, a skill developed during sequential learning. Conversely, prolonged training with sequential signals can weaken the model's capacity to produce parametric sequences that render visually natural objects, a capability enhanced during the visual feedback stage. To mitigate this, we introduce an alternate training strategy (Figure 2(c)). The process begins with the sequential learning stage, ensuring the model acquires the ability to generate logically coherent parametric sequences. Subsequently, the training is divided into smaller blocks. Within each block, the model first learns from the visual signal, followed by the sequential signal, balancing the two objectives effectively.

## 4. Experiments

### 4.1. Setups

**Datasets.** For the dataset used in the sequential learning stage, we use DeepCAD dataset (Wu et al., 2021) as the source for CAD parametric sequences (specifically the version processed by Xu et al. (2022)). We construct a dataset compromising 20k pairs of textual instructions and CAD parametric sequence using the techniques introduced in Section 3.2 and Appendix B.3. For the preference data used in the visual feedback stage, we employ `llava-onevision-qwen2-7b` (Li et al., 2024a) to

construct it using the method introduced in Section 3.3. For each iteration of the visual feedback, we generate approximately 1,500 preference pairs, by using 1,000 text prompts as input, sampling 5 times per prompt, and filtering out invalid or low-quality samples. For the test set, we construct it by splitting the dataset used in sequential learning into train, validation, and test sets with a 90:5:5 ratio.

**Implementation Details.** `LLaMA-3-8b-Instruct` is used as the LLM backbone, with a maximum token length of 1024. For efficient fine-tuning, we adopt Low-Rank Adaptation (LoRA) (Hu et al., 2022) with hyperparameters $r = 32$ and $\alpha = 32$. The initial sequential learning stage lasts for 40 epochs with a learning rate of $1 \times 10^{-4}$, using the AdamW optimizer. Following this, we run 5 iterations of alternating visual feedback and sequential learning stages. The visual feedback stage lasts for 5 epochs on the preference data, while the sequential learning stage lasts for 1 epoch using the same dataset as the initial sequential learning stage. Training is conducted on four NVIDIA A6000-48GB SMX GPUs using PyTorch Distributed Data Parallel (DDP).

**Baselines.** We consider two types of baselines. The first is a specialized model for Text-to-CAD (Khan et al., 2024b; Li et al., 2024b). We use Khan et al. (2024b) as our baseline, as Li et al. (2024b) is not open-sourced and we were unable to reproduce it ourselves. The second baseline is a general model that acquires some CAD knowledge during pre-training. We use the most powerful model, GPT-4o, as our baseline. Specifically, we apply few-shot learning, providing 8 examples as input for GPT-4o.

**Metrics.** Our evaluation focuses on assessing the alignment of generated CAD models with input instructions and the overall quality of the generated CAD models. We employ the metrics at both the sequential level and visual level. First, to evaluate the correspondence between the ground-truth and generated parametric sequences, we use F1 scores following Khan et al. (2024b). Specifically, we compute F1 score for primitives (averaged over lines, arcs, and circles for brevity) and extrusions, denoted as **F1-Sketch** and **F1-Extrusion**. Second, to assess the quality of the generated CAD models, we compare the ground-truth and generated point clouds. We adopt Chamfer Distance (**CD**) from Khan et al. (2024b) and additional metrics from Xu et al. (2022), including Coverage (**COV**), which quantifies the percentage of real data covered by generated samples using CD; Minimum Matching Distance (**MMD**), which evaluates the closest match between generated samples and real data; and Jensen-Shannon Divergence (**JSD**), which measures distribution similarity. Additionally, we compute the Invalidity Ratio (**IR**), which quantifies the percentage of generated parametric sequences that fail to render into valid visual objects. Furthermore, we introduce an LVM-based metric, denoted as **LVM Score**, to assess the visual correspondence between model predictions and input instructions. To this end, we employ `GPT-4o` with a dedicated evaluation prompt. Further details are provided in Appendix D.1. Finally, we conduct human assessments to rank generations from different baselines, denoted as **Avg. Rank**. Details on this evaluation can be found in Appendix D.2.

## 4.2. Main Results

**Quantitative Evaluation.** Table 1 summarizes the quantitative results comparing CADFusion with baseline methods (see Appendix D.4 for more details). Compared to `GPT-4o`, CADFusion outperforms it across all metrics. This suggests that while the general model may have acquired some CAD knowledge during pre-training, explicitly optimizing for Text-to-CAD, as in our approach, is crucial for improving performance, Compared to Text2CAD[2], CADFusion achieves comparable or better performance on all metrics, with particular strengths in metrics reflecting the visual quality such as LVM score and Avg. Rank. This highlights the effectiveness of incorporating visual signals in our approach, as opposed to Text2CAD, which relies solely on sequential signals. This outcome also aligns with Khan et al. (2024b)'s limitation statement that Text2CAD is limited to generating only rectangular and cylindrical shapes. When faced with complex geometries, it struggles to perform effectively.

**Qualitative Evaluation.** Figure 5 compares the results among the ground truth, our method, GPT-4o, and Text2CAD on the test set. GPT-4o frequently fails to produce renderable results across most test cases, which aligns with its high invalidity ratio (IR) reported in Table 1. While it occasionally generates valid shapes, its outputs are often misaligned with the input prompts. Text2CAD generate well-formed shapes without irregular edges or corners. However, it often produces oversimplified shapes and, for more complex prompts, tends to generate multiple cubes or panels instead of accurately capturing the intended structure. This aligns with its low invalidity ratio (IR) but poor visual scores. such as LVM score and Avg. Rank, in Table 1. CADFusion provides the most precise response to input instructions and achieves the highest similarity to the ground truth. It successfully captures complex shapes, including rectangles, hexagons, and nested structures, such as a hexagonal hole within a cylinder. Additionally, it exhibits a strong understanding of language cues, accurately interpreting numerical and qualitative descriptors like "long" or "T-shape". Additional qualitative results, as well as our model's ability to generate multiple varied outputs, are presented and discussed in Appendix D.6 and D.7.

---

[2]Our comparison with Text2CAD is not entirely aligned and is in favor of it. Performing poorly on prompts we provided, we have present the results of Text2CAD tested with their original prompts. We detail this problem in Appendix D.4.

| Method | F1↑ | | CD↓ | COV ↑ | MMD ↓ | JSD ↓ | IR ↓ | LVM Score ↑ | Avg. Rank ↓ |
|---|---|---|---|---|---|---|---|---|---|
| | Sketch | Extrusion | | | | | | | |
| **GPT-4o** | 82.96 | 85.72 | 68.50 | 72.40 | 6.60 | 37.93 | 74.26 | 5.13 | 3.22 |
| **Text2CAD** | 63.94 | 92.13 | 30.23 | - | - | - | **3.37** | 2.01 | 2.97 |
| **CADFusion** | **85.22** | **92.79** | **19.89** | **90.40** | **3.49** | **17.11** | 6.20 | **8.96** | **1.86** |

*Table 1.* Quantative results - Test results on F1 scores including **Sketch** (primitive, averaged) and **Extrusion**, Chamfer Distance (**CD**), Coverage (**COV**), Minimum Matching Distance (**MMD**), Jensen-Shannon Divergence (**JSD**), Invalidity Ratio (**IR**), the **LVM Score** and the average rank from human evaluation (**Avg. Rank**). An upward arrow (↑) indicates that higher values are better, while a downward arrow (↓) signifies that lower values are preferred. Since Text2CAD does not release COV, MMD, and JSD, and we were unable to compute them ourselves due to differences in setup, these values are unavailable.

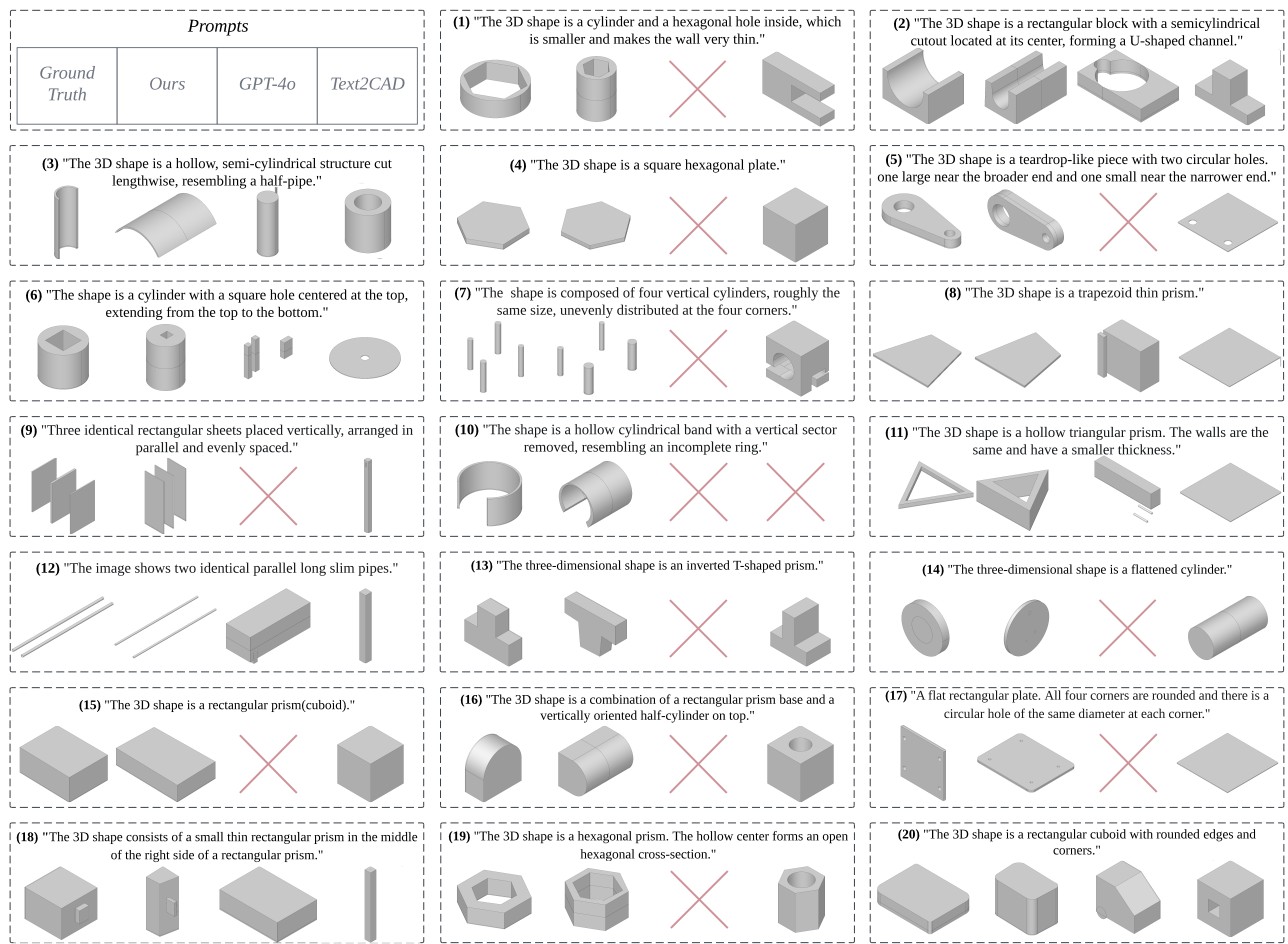

*Figure 5.* Qualitative results. The input prompt is shown at the top of each subsection. Images are arranged from left to right in the following order: ground truth, CADFusion, GPT-4o, and Text2CAD. Outputs that cannot be rendered are marked with a red cross. CADFusion outperforms all baselines in understanding instructions and generating CAD objects that are both sequentially and visually high quality. GPT-4o frequently produces invalid samples and pays little attention to shape details. Text2CAD generates well-formed basic shapes with a regular appearance but struggles to accurately follow input instructions and represent complex geometries.

### 4.3. Ablation Studies

We conduct ablation studies on the effectiveness of the visual feedback stage, the impact of the alternate training, and the choice between human and LVM annotation for data.

**Visual Feedback.** To assess the importance of visual feedback, we conduct an ablation study on CADFusion using only sequential learning, denoted as CADFusion$_{SL}$. The first row of Table 2 presents its LVM score and invalidity ratio. Compared to our approach, denoted as CADFusion$_{SL\text{-}VFSL(5)}$ in Table 2, while CADFusion$_{SL}$ improves the invalidity ratio by 1.36%, it results in a considerable decrease in the

| Method | LVM Score ↑ | IR ↓ |
|---|---|---|
| CADFusion$_{SL}$ | 7.69 | 4.84 |
| CADFusion$_{SL_{w/o HA}}$ | 6.56 | 6.00 |
| CADFusion$_{SL-VF}$ | 5.94 | 88.87 |
| CADFusion$_{SL-VF_{RPO}}$ | 6.21 | 3.46 |
| CADFusion$_{SL-VFSL(1)_{w/ HA}}$ | 8.28 | 17.03 |
| CADFusion$_{SL-VFSL(1)}$ | 8.76 | 4.42 |
| CADFusion$_{SL-VFSL(3)}$ | 8.89 | 4.21 |
| CADFusion$_{SL-VFSL(5)}$ | 8.96 | 6.20 |

*Table 2.* LVM scores and invalidity ratios across different CADFusion variants. The suffix SL indicates that the model is trained with the initial Sequential Learning stage, while VF denotes the Visual Feedback stage without additional Sequential Learning. VFSL represents Visual Feedback with alternating Sequential Learning. The tag w/ HA signifies that the data is preprocessed with human annotation, whereas w/o HA denotes the absence of human annotation. Numbers in parentheses indicate the number of VFSL rounds performed. RPO refers to the model using Regularized Preference Optimization (RPO) (Liu et al., 2024) to stabilize DPO.

LVM score. This underscores the crucial role of the visual feedback stage: by leveraging visual preference data, our framework effectively enhances the visual quality of the generated CAD models. Additionally, CADFusion$_{SL}$ outperforms the baseline method, Text2CAD, which also relies solely on sequential signals. Note that this advantage is achieved using 20k data, while Text2CAD uses 150k data. This demonstrates the effectiveness of the techniques employed in our sequential learning stage, including leveraging LLMs as the backbone, representing CAD parametric sequences as textual tokens, and utilizing human annotations (Section 3.2).

**Alternate Training.** In Section 3.4, we propose an alternate training strategy to retain the benefits of both sequential learning and visual feedback stage. We compare this approach with three variations: 1) visual feedback only (CADFusion$_{SL-VF}$), 2) visual feedback with an additional Negative Log Likelihood loss (CADFusion$_{SL-VF_{RPO}}$) to regularize and stabilize DPO (Liu et al., 2024), and 3) iterative visual-sequential training (our method).

Table 2 presents the results, with our approach denoted as CADFusion$_{SL-VFSL(5)}$. The high invalidity ratio of CADFusion$_{SL-VF}$ indicates that it struggles to generate renderable sequences, suggesting that extended training with visual signals can impair the model's ability to generate well-formatted parametric sequences. Besides, CADFusion$_{SL-VF}$ receives a low rating from the LVM judge, revealing that training with visual feedback along provides limited benefit. Regarding CADFusion$_{SL-VF_{RPO}}$ which incorporates the additional loss, while it achieves low invalidity ratio, its visual quality, as assessed by the LVM judge, is even lower than the SL-only setup (i.e., CADFusion$_{SL}$). This indicates

that it fails to effectively balance the contributions of both sequential signals and visual signals.

We also compare model variants that use different numbers of iterations of visual feedback and sequential learning. In Table 2, for each CADFusion$_{SL-VFSL(*)}$ variant, the number in parentheses indicates the number of alternative training rounds performed. The results for iterations 1, 3, and 5 are reported, showing a gradual increase in LVM scores along with a stable invalidity ratio. This further validates the effectiveness of our approach.

**Data Annotation.** We examine the impact of our choice of data annotation. In the sequential learning stage, the dataset is constructed by first using LVMs to generate initial captions, followed by human annotators refining them. To evaluate the effect of this decision, we conduct an experiment in which our method is trained on data without human annotation, denoted as CADFusion$_{SL_{w/o HA}}$. The second row of Table 2 presents the results. It shows worse LVM score and IR compared to the version using data with human annotations (CADFusion$_{SL}$), highlighting the necessity of human annotation in the sequential learning stage.

In the visual feedback stage, LVMs are used to score CAD models and generate preference data. This design choice is driven by the high cost of human annotation and the challenge of managing human annotators to ensure consistent scoring. To evaluate the effect of this decision, we conduct an experiment where the visual feedback stage of our method is trained on human-scored preference pairs, denoted as CADFusion$_{SL-VFSL(1)_{w/ HA}}$. Compared to the LVM-scored version (i.e., CADFusion$_{SL-VFSL(1)}$), it achieves a worse LVM score and IR. This aligns with our intuition that, while human annotation may be more accurate, managing annotators for consistent scoring is difficult. Furthermore, using LVM-scored preference data allows CADFusion$_{SL-VFSL(1)}$ to scale across more rounds of visual feedback (e.g., CADFusion$_{SL-VFSL(5)}$), leading to improved performance. Achieving this with human annotation would be challenging and expensive.

## 5. Limitation

CADFusion's results are overall promising. However, there are limitations that could be addressed in future work. First, modern LVMs suffer from performance drop when handling multiple images as input. Currently, we can only provide LVM with a single-view image to ensure both accurate image understanding and prompt following. This limitation prevents us from achieving a more effective Visual Feedback pipeline and evaluator. Second, CADFusion struggles to generate very complex shapes that require spatial and commonsense reasoning, such as the shapes of letters and words (see Appendix D.8).

# 6. Conclusion

We propose CADFusion for Text-to-CAD, the first approach to incorporate visual feedback from rendered CAD objects into the training pipeline. CADFusion uses LLMs as backbone and alternates between the sequential learning stage and the visual feedback stage. We conduct extensive experiments to demonstrate the superiority of CADFusion and validate the effectiveness of the design choices. In the future, we plan to further improve the preference data construction pipeline to enhance performance, and collect more CAD data with more complex geometric shapes to investigate CADFusion's performance on more challenging cases.

# Acknowledgement

We thank Weijian Ma for insightful conversation and discussion. We also appreciate the informative comments and suggestions provided by the anonymous reviewers.

# Impact Statement

This paper presents work aimed at improving Text-to-CAD generation through the use of LLM-based frameworks and the incorporation of visual feedback. Our work has the potential to enhance the CAD design process, offering the benefits of automation and efficiency while reducing reliance on intensive training and specialized expertise. This could make CAD design more accessible, particularly in industries where skilled designers are in short supply or where rapid prototyping is essential.

# Ethics Statement

In this work, we have invited crowd workers to give textual descriptions to CAD models. We conducted this work in accordance with ethical guidelines to ensure that participants were treated fairly, respectfully, and safely throughout the process. We took steps to protect the privacy of crowd workers by not collecting personally identifiable information. The data annotated by the crowd workers was used only for research purpose related to improving CAD generating techniques.

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

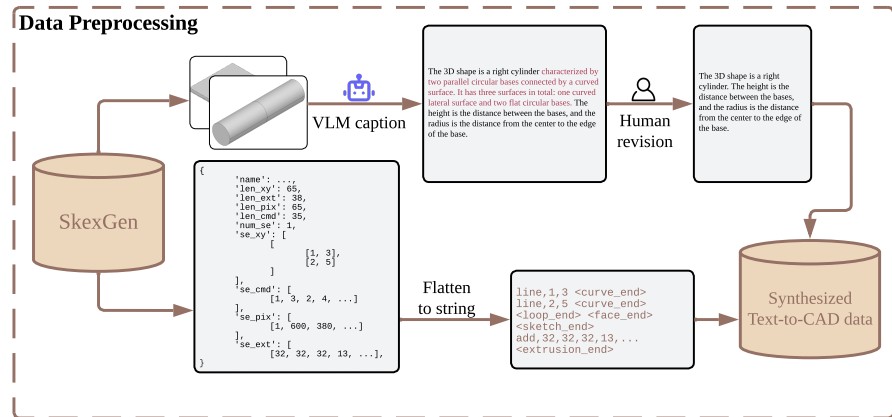

*Figure 6.* An overview of the data preprocessing steps. The original dataset is transformed into captions that serve as textual inputs, while the corresponding stringified CAD representations are used as ground truth references.

## A. User Guidelines for Prompting

A good prompt follows a structured description: (1) *shape overview*, (2) *shape details*, and (3) *shape applications*. Given the varying shape complexity, we encourage but do not enforce describing item (2) and (3). Below is an example caption retrieved from Figure 8, Row 2, Item 3, demonstrating this approach:

```
"""
[Shape Overview] The 3D shape consists of a large, flat rectangular slab with two evenly
spaced, identical cylindrical protrusions extending vertically from its surface. [Shape
Details (Optional)] The slab provides a stable base with significant length and width
compared to its thin height, while the cylinders are relatively short and have small
diameters. [Shape Applications (Optional)] The overall design is symmetrical and balanced,
    potentially serving as a mounting base or connector.
"""
```

*Listing 1.* User guidelines for prompting.

## B. Additional Dataset Construction Detail

### B.1. Converting Raw Data into Strings

**CADFusion's String Format** Our representation adopts the Sketch-and-Extrude Modeling (SEM) format, wherein a CAD instance is composed of sketches and extrusions. Each sketch is structured into multiple faces, and each face comprises multiple loops. Within each loop, geometric primitives such as lines, arcs, and circles are parameterized as follows:

- **Line**: Represented by a line identifier and one coordinate.
- **Arc**: Defined by an arc identifier and two coordinates.[3]
- **Circle**: Represented by a circle identifier and four coordinates.

Each extrusion is represented as a sequence formatted as `BVVTTTRRRRRRRRRRSOO`, where the components are defined as follows:

- `B`: The boolean operation, selected from `add`, `cut`, `intersect`.

---

[3]While lines and arcs generally require 2 and 3 coordinates for representation, respectively, this work leverages a simplified representation where the endpoints of lines and arcs are determined by the first point of the subsequent curve. If the loop is closed at the current curve, its endpoint is determined by the first curve in the loop.

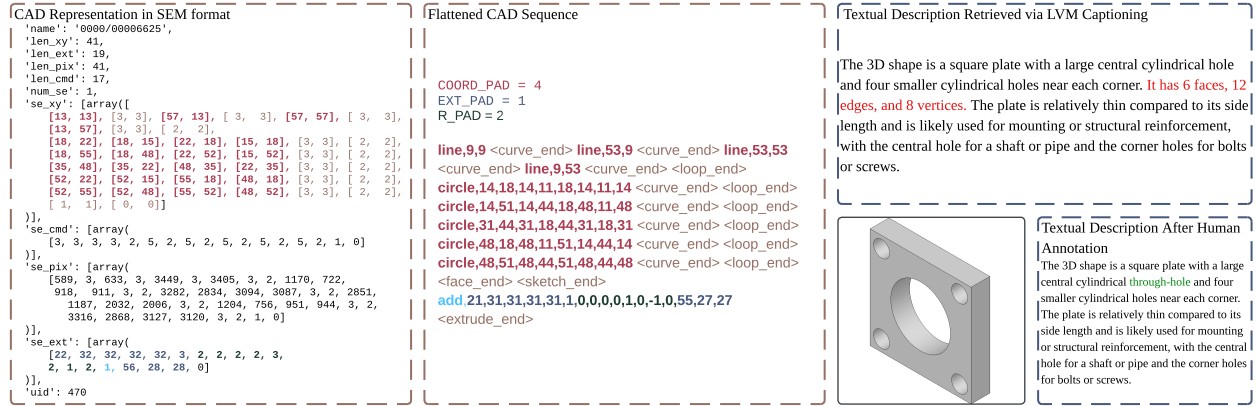

*Figure 7.* An overview of multiple CAD representations and their corresponding captions. Left: A CAD representation in the raw SEM format alongside its stringified sequence, with values highlighted in different colors based on the padding used for decoding. Right: Captions generated by the LVM and refined by human annotation. Phrases removed during human fine-tuning are marked in red, while those added by humans are marked in green. All representations and captions correspond to the same CAD figure, which is displayed in the bottom-right corner.

- V: The displacements of the top and bottom planes from the reference plane.

- T: The 3D translation vector.

- R: The 3D rotation, represented as a quaternion or equivalent.

- S: The scaling factor.

- O: The center of scaling.

**Converting Source Data to CADFusion's Format**  The original representation is derived from the SkexGen dataset (Xu et al., 2022). Each CAD instance includes several components: sketch commands, sketch coordinates, and extrusion commands, which are stored in the se_cmd, se_xy, and se_ext entries, respectively. The lengths of these entries correspond to the number of sketch-extrusion pairs within the complete CAD shape. To convert these entries into strings, we iteratively describe the sketches and extrusions in our format, ensuring that the resulting sequence reflects the chronological design order of the CAD process.

In iteration $i$, we select the $i$-th item from the se_xy, se_cmd, and se_ext entries. For each digit in the se_cmd array, we perform operations based on the command value as follows:

- **Command value** = 5: Create a circle; use the first 4 items in se_xy as XY coordinates and append the <curve_end> token. Skip 5 positions in the se_xy array.

- **Command value** = 4: Create an arc; use the first 2 items in se_xy as XY coordinates and append the <curve_end> token. Skip 3 positions in the se_xy array.

- **Command value** = 3: Create a line; use the first item in se_xy as an XY coordinate and append the <curve_end> token. Skip 2 positions in the se_xy array.

- **Command value** = 2: Mark the end of the loop by appending the <loop_end> token. Skip 1 position in the se_xy array.

- **Command value** = 1: Mark the end of the face by appending the <face_end> token. Skip 1 position in the se_xy array.

- **Command value** = 0: Mark the end of the sketch by appending the <sketch_end> token. Skip 1 position in the se_xy array.

Extrusions are represented by the 1D array `se_ext`. The operation identifier is translated into a word, and the remaining values are flattened. To distinguish coordinates from special tokens, all coordinates are initially padded; they are subsequently unpadded based on the original padding values. Figure 7 illustrates the conversion process from the SkexGen representation format to our stringified sequence.

### B.2. Generating Textual Instructions

Textual instructions are generated in two steps: first, by applying LVM captioning on single-view images of CAD models; second, through human refinement of the generated captions to ensure clarity and accuracy.

Given a sequence representation, the CAD instance is rendered into an image, and captions are generated using GPT-4o. This step is designed to extract geometric properties, including the number of shapes, their dimensions, spatial arrangements, and other relevant details. The prompt used for this step is provided in Listing 1.

```
1  {
2      "Prompt1": "Propose a series of questions about the 3D shape and give the answers. The
        first question should ask for a detailed description and others should focus on the
        specific geometric properties, number, size proportions and positional relationship,
        and other details.",
3      "Prompt2": "Based on the dialogue, please give a final description of the 3D shape. No
        more than 70 words."
4  }
```

*Listing 2.* Prompts that are used for making captions. The first prompt is used to generate question-answer pairs, and the second prompt collects and summarizes the informations in the first prompt to yield the final caption.

The LVM-generated captions are further refined by human annotators to produce fine-grained captions that can serve as precise textual instructions. The human annotators follow these guidelines during the editing process:

- **Ensuring Correspondence**: The description must accurately reflect the figure without any discrepancies.

- **Ensuring Succinctness**: The description should be as concise as possible while maintaining clarity and completeness.

- **Permission for Removal**: Figures that are excessively complex or challenging to describe may be excluded from the dataset. In practice, the annotators are permitted to mark the revised descriptions of such instances as "null".

Figure 7 illustrates an example of how an image is captioned by the LVM and subsequently refined by human annotators.

### B.3. Dataset Construction

The dataset construction process is illustrated in Figure 6. Starting with a CAD representation from the original dataset, we generate a paired textual instruction and a stringified CAD representation. The textual instruction is created through the captioning process detailed in Section B.2, while the ground truth reference is obtained by converting the CAD formatting as described in Appendix B.1.

## C. Additional Training Detail

### C.1. Sequential Learning

We fine-tune a `LLaMA-3-8b-Instruct` by 40 epochs on 4 NVIDIA A6000-48GB SMX GPUs with a LoRA with rank 32. Further details regarding the fine-tuning process are provided in the Experiment Section of the main paper. The specific prompt used for the learning is as follows:

```
1      "Below is a description of a 3D shape:\n
2      {description}\n
3      Generate a Computer-Aided Design (CAD) command sequence of the 3D shape:\n"
```

*Listing 3.* Prompt used for sequential learning. description refers to the actual textual commands of samples

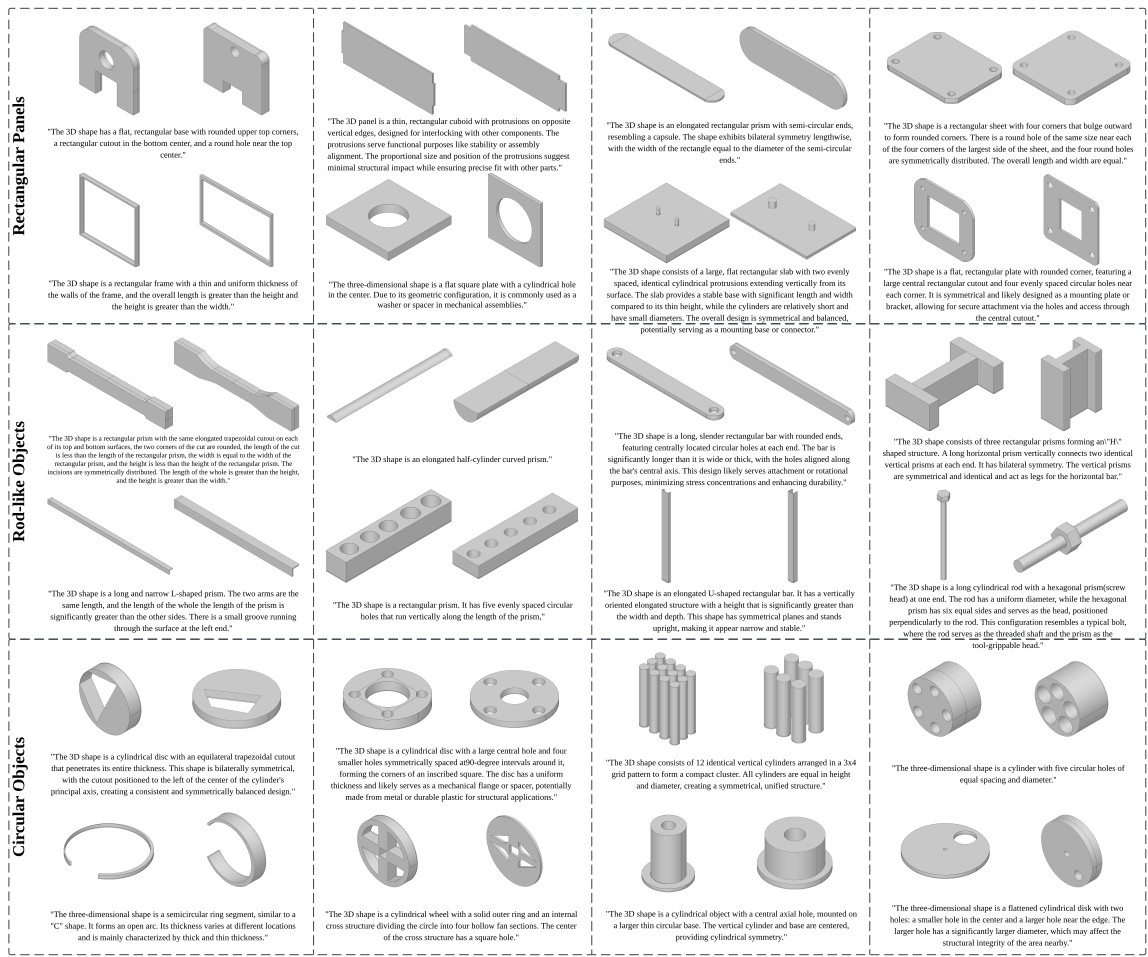

*Figure 8.* Additional qualitative results, Part 1. The results are grouped by categories such as panels and circular objects. In each sub-figure, the left image shows the figure rendered from the ground truth, while the right image displays the generation by CADFusion. The corresponding textual instructions are provided at the bottom.

## C.2. Visual Feedback

The visual feedback is collected as outlined in the main paper. The `llava-onevision-qwen2-7b-ov-chat` model is utilized to generate visual descriptions. For each input sequence produced by the post-SL CADFusion, the corresponding rendered figure is evaluated by the model, which assigns a score ranging from 0 to 10. The prompt used for this scoring process is detailed in Listing 3:

```
1    "You are a harsh grader for new CAD  designers' works. The following is a text
     description of a CAD figure that they designed and an image of a CAD instance. \n
2    Description: {description} \n
3    Comment on this work for \n
4    1. If the overall shape remains correct; \n
5    2. If the number of components are correct, especially the circular holes; \n
6    3. If the distribution of the components are natural, i.e. they are not clustered
     together or collide with each other. \n
7    After that, give a score out of 10. Do not comment on issues such as texture,
     smoothness and colors."
```

*Listing 4.* Prompt used by LLaVA-OV for scoring an input figure. description refers to the textual of the sample.

The DPO procedure is conducted on 4 NVIDIA A6000-48GB SMX GPUs with a LoRA with rank 32. The training involves

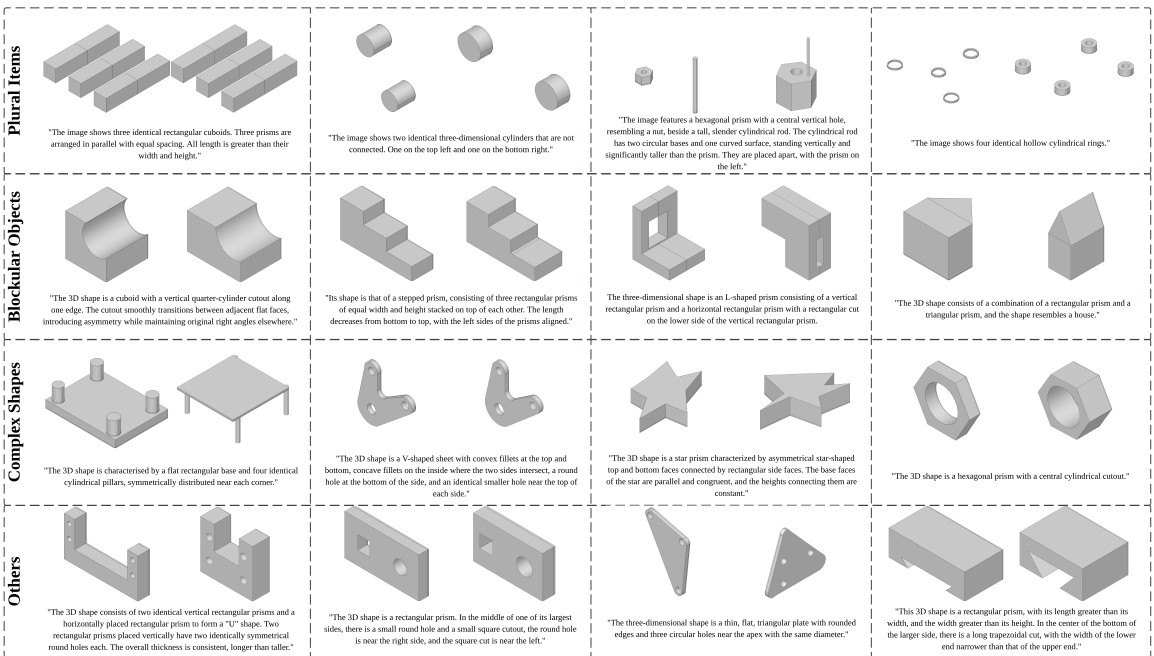

*Figure 9.* Additional qualitative results, Part 2. The results are grouped by categories such as multiple distinct items and complex shapes. In each sub-figure, the left image shows the figure rendered from the ground truth, while the right image displays the generation by CADFusion. The corresponding textual instructions are provided at the bottom.

five iterative DPO/SFT rounds, which require approximately 2.5 days to complete.

## D. Additional Experimental Results

### D.1. LVM Evaluation Setups

As mentioned in the main paper, we used a `GPT-4o` model as the LVM evaluator. It is selected over `LLaMA-ov` because we attempt to prevent the impact of AI bias that makes it prefer its own generation [4]. The prompt we used for LVM evaluation is detailed in Listing 4:

```
"
The following is a text description of a 3D CAD figure and an image of a CAD instance.
    Measure if the figure corresponds to the given description, and give a score in the
    scale of 10. Do not comment on issues such as texture, smoothness and colors \n
    description: {description}\n
"
```

*Listing 5.* Prompt used by GPT-4o for evaluation.

### D.2. Human Evaluation Setups

We generate a quadruple of outputs for each test set instruction. Each quadruple presents four rendered generations from CADFusion$_{SL}$, CADFusion, GPT-4o and Text2CAD, respectively. The generations are tested by their correspondence between CAD shapes and instructions. Six human judges (with college-level or higher education records) are asked to rank the generations[5] with the first place being the best model. The ranks are then collected and averaged to be the scores we

---

[4]We acknowledge that this choice may introduce bias in the GPT-4o results. However, we have decided to proceed with it for two reasons: 1) our primary focus is on comparing Text2CAD with our model, and 2) the GPT-based generations during our experiment showed a big margin in LVM scores compared to other methods, so the impact of this bias is minimal.

[5]Due to the lack of overlapping, we obtain approximately 50 unique samples.

presented as human evaluation.

### D.3. GPT Baselines

The prompt we used for the GPT-4o baseline is detailed in Listing 5.

```
1  "
2      Below is a description of a 3D shape:
3      {description}
4      Generate a Computer-Aided Design (CAD) command sequence of the 3D shape. The command
       sequence involves sketches such as lines, arcs, and circles, each marked by the
       endpoints, and extrusions that make the sketch into 3D volumes.
5
6      Here are some examples:
7      1. <few shot example>
8      2. <few shot example>
9      3. <few shot example>
10     4. <few shot example>
11     5. <few shot example>
12     6. <few shot example>
13     7. <few shot example>
14     8. <few shot example>
15
16     Now it's your turn. Remind that this is your description: {description}. No
       explanation is needed. Only return your final sequence, and in one line.
17 "
```

*Listing 6.* Prompt used by GPT-4o for baseline comparison.

Alongside the current 8-shot version, we also tested a 3-shot GPT-4o model to reduce computation costs. However, the 3-shot model resulted in approximately a 92% invalidity ratio, and the 8% of renderable outputs were barely recognizable in relation to the prompt. Given these issues, we have decided to use the 8-shot version as our baseline for GPT.

### D.4. Additional Statements on Text2CAD Results

In the quantitative experiments, our setups are not fully aligned with those of Text2CAD. This discrepancy arises because when we used our test set prompts as input, we observed a performance degradation and a significant gap between our computed results and those reported by the authors.

We discovered that the discrepancy stems from the model's sensitivity to the level of detail in the prompt. Text2CAD performs well only with expert-level prompts, which contain step-by-step sketching guidelines. Our prompts, however, do not include this level of detail [6]. To ensure consistency, in Table 1, we report Text2CAD's performance based on their expert prompts when computing the metrics they introduced. Specifically, for each item in the test set, CADFusion and GPT-4o's predictions were generated using our prompts, while Text2CAD's predictions were generated using the expert prompt for the same item from their prompt base.

This approach aligns the results we reproduced with the reported scores from the original paper. To present a comprehensive and accurate study, we also report Text2CAD's results using our prompts and intermediate-level prompts in Table 3. The last two rows, CADFusion and Text2CAD-our-prompt, are aligned as the same prompt is used.

By changing the prompt from the expert-level prompt in their database to an intermediate-level prompt, we observe a similar performance drop. This indicates that our prompting method does not degrade Text2CAD's performance. Instead, it is an limitation stemmed from Text2CAD itself. Our model, using a simplified prompt, outperforms Text2CAD-expert. Given that the expert-level prompt from Text2CAD is too long and too specific to be feasible in the real designing process, we believe that our quantitative advantage over it is non-trivial.

---

[6]We are concerned that the impact of detailed prompts containing step-by-step instructions and point coordinates is limited, as they may not be feasible in real-life scenarios.

| Method | F1↑ | | | | CD↓ |
|---|---|---|---|---|---|
| | Line | Arc | Circle | Extrusion | |
| **Text2CAD-intermediate** | 66.65 | 4.85 | 47.62 | **93.56** | 146.15 |
| **Text2CAD-expert** | 79.59 | 42.79 | 69.45 | 92.13 | 30.23 |
| **Text2CAD-our-prompt** | 54.42 | 0.92 | 18.42 | 75.37 | 235.91 |
| **CADFusion** | **83.71** | **81.99** | **89.97** | 92.79 | **19.89** |

*Table 3.* Results of our model and different Text2CAD prompts on metrics Proposed by Khan et al. (2024b). The suffix indicates the prompt type used for testing. Text2CAD-ours and CADFusion are the most aligned pairs, while Text2CAD-expert and CADFusion are the ones reported in the main paper.

| Method | LVM Score ↑ | IR ↓ |
|---|---|---|
| CADFusion$_{SL}$ | 7.69 | 4.84 |
| CADFusion$_{SL\,w/o\,HA\sim18k}$ | 6.56 | 6.00 |
| CADFusion$_{SL\,w/o\,HA\sim170k}$ | 6.60 | 9.04 |

*Table 4.* LVM scores and invalidity ratios across different CADFusion variants. All three models are trained using only the initial Sequential Learning stage. The suffix `w/o HA` indicates that the variant does not use human-annotated data, while the number denotes the size of the training set.

| Method | Avg. Rank ↓ |
|---|---|
| **GPT-4o** *-8shot* | 3.22 |
| **Text2CAD** | 2.97 |
| **CADFusion**-*SFT only* | 2.03 |
| **CADFusion** | 1.86 |

*Table 5.* Human Evaluation Results. Human annotators ranked the generations of different methods based on their quality, with a lower rank indicating higher human preference.

### D.5. Additional Quantative Results

We report additional quantitative results in this section.

**On Dataset Size.** The dataset used in our experiments is a subset of SkexGen (Xu et al., 2022). Since human annotation is not scalable, we evaluate the trade-off between scalability and data quality. One such evaluation, discussed in the Ablation Study (Section 4.3), demonstrates that, given the same number of training samples, data quality outweighs dataset scalability in terms of model performance.

Additionally, we investigate whether increasing dataset size can mitigate quality limitations by conducting an experiment on the full SkexGen-based Text-to-CAD dataset ( 170k samples). The results, presented in Table 4, indicate that increasing dataset size does not significantly improve the visual quality of model generations. While a slight performance gain is observed with additional training samples, the improvement is marginal, and none of the w/o HA variants outperform the human-annotated counterpart.

**On Human Evaluation.** We conducted human evaluations across four models: GPT-4o, Text2CAD, CADFusion, and CADFusion trained only with the Sequential Learning stage. However, only the first three models are reported in Table 1. The complete results of human evaluation are presented in Table 5. As indicated by the evaluation, the two CADFusion variants are preferred over the baselines, with the version incorporating Visual Feedback receiving higher rankings from human judges. This highlights the effectiveness of visual feedback in improving model performance.

### D.6. Additional Qualitative Results

In this section, we present additional qualitative results. Figures 8 and 9 display these results, organized by CAD shape properties such as panels and circular objects. These examples demonstrate that our model can efficiently handle a variety of CAD shapes with distinct instructions, such as holes and frames. Furthermore, the model performs well in generating multiple identical objects, as shown in the first row of Figure 9, and can effectively generate more complex shapes, such as stars and V-shapes.

### D.7. Text to Multiple CAD Figures

During inference, we set the temperature $t = 0.3$, top_p $= 0.9$, and top_k $= 50$ to enable non-deterministic generation. This configuration allows us to produce varied CAD figures that meet the instructed requirements, with slight differences between them. As a result, users can select the design that best aligns with their specific needs. Examples of such outputs are shown in Figure 10. These results demonstrate that while adhering to the provided instructions, CADFusion is capable

The 3D shape is a rectangular prism with the same elongated trapezoidal cutout on each of its top and bottom surfaces, the two corners of the cut are rounded, the length of the cut is less than the length of the rectangular prism, the width is equal to the width of the rectangular prism, and the height is less than the height of the rectangular prism. The incisions are symmetrically distributed.

The three-dimensional shape is a semicircular ring segment, similar to a "C" shape. It forms an open arc. Its thickness varies at different locations and is mainly characterized by thick and thin thickness.

The 3D shape is a flat, rectangular plate with rounded corner, featuring a large central rectangular cutout and four evenly spaced circular holes near each corner. It is symmetrical and likely designed as a mounting plate or bracket, allowing for secure attachment via the holes and access through the central cutout.

Its shape is that of a stepped prism, consisting of three rectangular prisms of equal width and height stacked on top of each other. The length decreases from bottom to top, with the left sides of the prisms aligned.

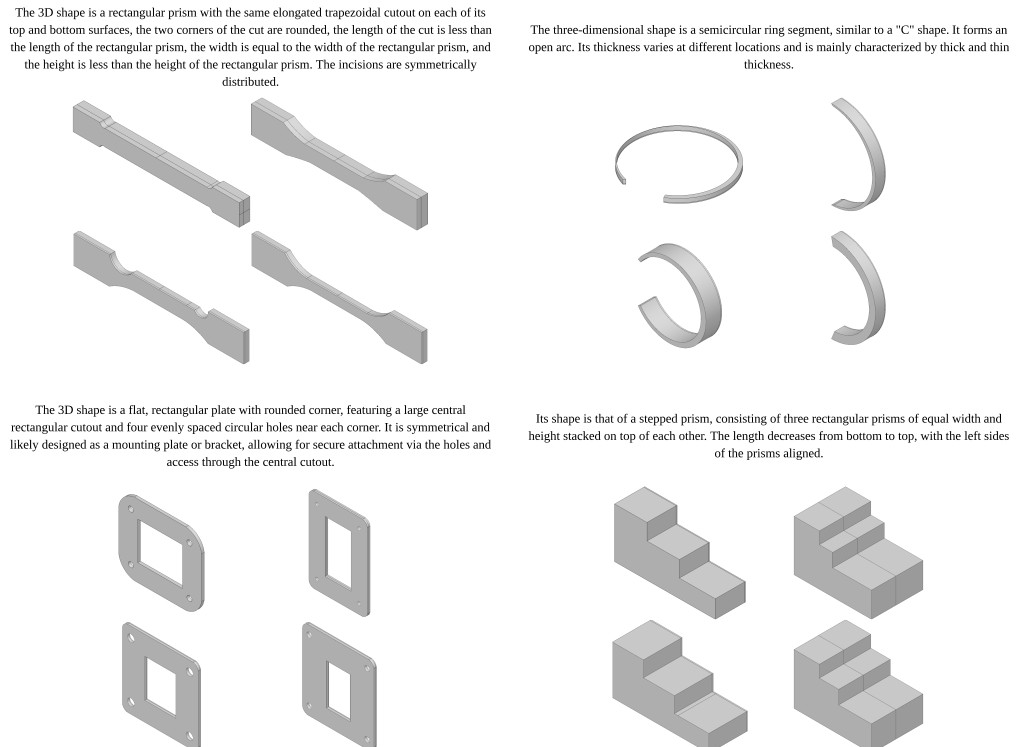

*Figure 10.* An overview of the generation of CAD instances with slight variations from a single prompt. In each sub-figure, the top-left image shows the ground truth generation, while the remaining three represent CADFusion's outputs, which exhibit variations in thickness, width, and cutout size. The prompt is displayed at the top of each sub-figure.

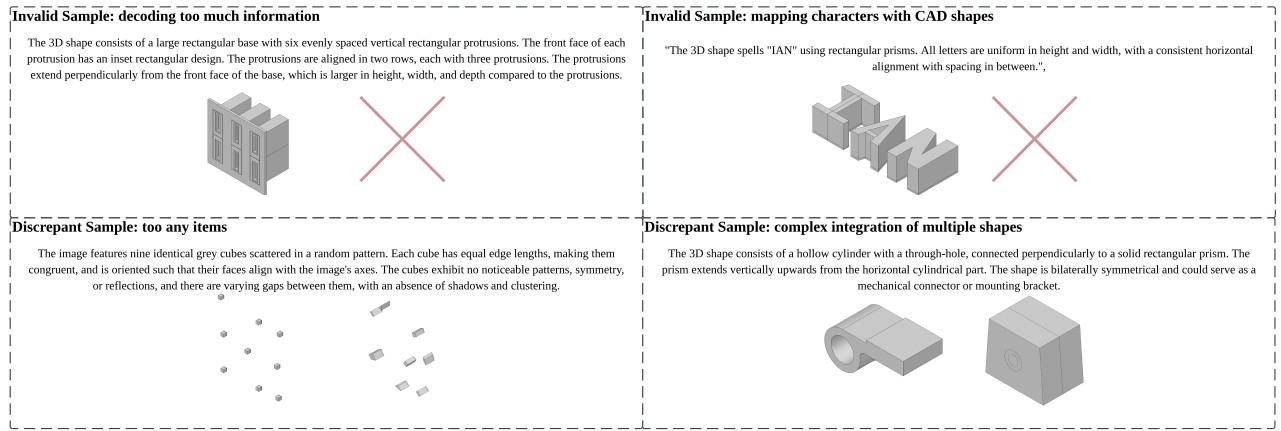

*Figure 11.* Invalid and discrepant samples. CADFusion generates invalid samples when the instructions are too complex or involve word shape knowledge, and produces discrepant outcomes when there are too many distinct items to generate or when complex merges are required to form the final CAD instance.

of generating diversified outputs. The variations primarily affect attributes such as thickness, width, and the size of holes and cutouts, while maintaining the overall shape. This flexibility offers users a broader range of choices, thereby reducing the amount of additional work required when integrating such Text-to-CAD systems into industrial applications.

## D.8. Failure Cases

We identify two types of failures in our work: sequences that are not renderable and shapes that are rendered but misaligned with the intended design. We refer to the former as **Invalid Samples** and the latter as **Discrepant Samples**. Examples of both types of failures are shown in Figure 11.

In our analysis, samples are often invalid when the input instruction is too complex, meaning there are too many elements to be drawn. The case shown in the top-left corner of Figure 11 involves more than 20 loops and 50 curves in the ground truth. Additionally, CADFusion struggles to map CAD shapes to characters such as letters, resulting in failures when attempting to construct shapes that spell words or names.

Discrepancies between the rendered shapes and the intended design can occur when the input instruction involves too many distinct items. While CADFusion demonstrates advanced capabilities in understanding numerical values compared to other models, handling more than 8 separate items remains a challenging task. In such cases, CADFusion may either miscalculate the number of items to draw or generate incorrect shapes, as shown in the bottom-left corner of Figure 11. Furthermore, integrating multiple items into complex shapes is another frequent challenge for CADFusion.

