# OpenReview forum: "Text-to-CAD Generation Through Infusing Visual Feedback in Large Language Models"
_ICML.cc/2025/Conference — ICML 2025 poster_

### Official Review · Reviewer_X4me · 2025-03-13

**Overall Recommendation:** 3

**Summary:**

The authors use DPO/Rinforcement Learning to fine-tune a LLM to produce CAD instructions from a prompt, so tht the rendered objects are ranked VLM. They fined tune Llamma and use prewarming to make sure it can generate CAD sewuences froma prompt before alternating betwen DPO and direct sequential learning steps. I beleive this is "On Policy DPO". The result is a model that scores higher than GPT-4o and one other method, according to  automatic and user-study based metrics on the quality of CAD files generated.

**Claims And Evidence:**

The paper lists 4 contributions - butthey dont seem tightly linked to the evaluation.
Working backwards from their evaluation, they are/shouldbe claiming:
1. Improved ability to handle simple prompts
2. Improved ability handle (more) complex shapes -- mainly shown in qualitative eval.
3. Higher accuracy / quality according to automatic benchmark and human scores
4. Human feedback in training improves quality
5. DPO (On policy) is effective for text-2-CAD to incorporate the appearance of the object.

These things are all evaluated at different levels, each with acceptable evaluation methods.

**Essential References Not Discussed:**

I think they should mention CAD-LLM and Vitruvian.

**Experimental Designs Or Analyses:**

As mentioned - I would like to know more about the user study.
I ampleased with the amount of qualitative results shown, and the metrics used.
There do not seem to be too-many baseliens to compare against in the acadmic literator (3D not 2D CAD, from text), and the wuthors were not able to implement one method, so they effectively compated only to Khan's work. I could not find a more appropriate reference either, although there is "zoo.dev" (commercial) and a text_to_cad github repo (no publication).  I think that what they have done is likely sufficient.

**Methods And Evaluation Criteria:**

The methods and evaluatin criteria make sense.
The authors use pre-existing benchmarks for the most part, introducing LVM score as a new metric.
This is a reasonable score as far as I can tell, and it is paired with more established ones.

Importantly, they include a user study.
The study had 6 people (I dont know who) rank 50 samples comparing 2 versions of CADFusion to the competing approachs (GPT-4o, Text2CAD).  One could complain that this is not enough, or that we do not know their qualifications (do they know anything about CAD?) or id they have potential bias -- was it a blind study where they dont know which is which?  A rubric or instructions to the participants would be nice. However I am pleased that the did a user study as I think it is important for this work.

**Other Comments Or Suggestions:**

L153 bad reference
L217 col2 - misplaced eqn number
not sure footnote 5 is needed.

**Other Strengths And Weaknesses:**

Strength:
- Interesting approach, nice way to handle non differentiabel rendering
- Good results qualitiatively and quant.

Weakness:
- I immediately found some refs that _seem_ relavant and I am concerned that they werent mentioned.
- Only comparing to one real other method (GPT-4o makes two, but it is not aimed at CAD)
- User study was small / underexplained

Neutral/Unsure:
- Relying on LLVM vs human feedback to evaluate

**Questions For Authors:**

How does your work relate to the CAD-LLM or Vitruvian work?
Can you explain your user-study a bit more? Was it blind? What were the users qualifications? What were their instructions?

**Relation To Broader Scientific Literature:**

I think there is a need for more editable structured representaitons from geenrative models and this is a step inthat direction. The visual feedback/DPO appeoach seems novel.

**Theoretical Claims:**

This is not a thoery paper -- there are not proofs or theortetical claims. They present a framework and use empirical evaluation with a small user study.

I would like to see discussion of static vs on-policy iterative DPO.

---

> ### Author Rebuttal · Authors · 2025-03-31
>
> Dear reviewer,
>
> Thank you for the valuable comments. We are pleased to know that you are happy with our evaluation details including the qualitative results and user study. We hope the following clarifications address your concerns.
>
> ## More References
> Thanks for pointing out these relevant works. Below is our discussion to them, and we will add them to our updated manuscript.
>
> 1. CADLLM [1]: It is a work on *unconditional* CAD generation. We referenced similar works such as DeepCAD [3] and SkexGen [4]. Our work is different to them by focusing on a *conditional* task that deals with text-guided CAD generation.
>
> 2. Vitruvion [2]: It focuses on generating 2D sketches from hand drawings. Our work, in contrast, does not focus on *CAD reconstructions* that takes visual inputs, but more abstract text inputs instead. We cited similar works to Vitruvion but on 3D reconstructions [5] [6].
>
> ## Baseline Methods
> As all reviewers acknowledge, the text-to-CAD task is relatively new - we have highlighted this in the introduction. As a result, there are only a few open-source baseline methods that are directly comparable to our approach. We sincerely thank you for your understanding on this matter.
>
> You are welcomed to have a look at our response to reviewer `3nTM`, in which we reported a comparison between our method with an adaption of Skexgen for text-based generation. We hope this confirms that our framework demonstrates superior capability among all existing works.
>
> ## Details of User Study
> Our user evaluation was conducted as a *blind* study. Participants were asked to **rank** the rendered CAD objects (without knowing their origin) based on 1. their **alignment** to the given textual descriptions, and 2. the **quality** of the CAD models. The users had college-level (or higher) education but we did not restrict their majors. Since our evaluation focuses solely on the visual appearance of the generated models, we believe this is appropriate.
>
> We have included an description of our human evaluation setup in Appendix C.2. However, we realize that this section may not be entirely clear and could be difficult to follow. We will update and clarify this section in the camera-ready version to provide more detailed instructions and improve overall readability.
>
> ## DPO: Static vs. On-policy
> Yes, our DPO implementation is not entirely "static". By splitting DPOs into smaller iterations and making it iterate with the sequential learning, the model being optimized is much closer to the policy model generating the sample pairs. It is a step towards an "on-policy" DPO.
>
> [1] Seff et al., Vitruvion: A Generative Model of Parametric CAD Sketches, ICLR 2022.
>
> [2] Wu et al., CAD-LLM: Large Language Model for CAD Generation, NeurIPS 2023 Workshop.
>
> [3] Wu et al., DeepCAD: A Deep Generative Network for Computer-Aided Design Models, ICCV 2021.
>
> [4] Xu et al., SkexGen: Autoregressive Generation of CAD Construction Sequences with Disentangled Codebooks, ICML 2022.
>
> [5] Ma et al., Draw Step by Step: Reconstructing CAD Construction Sequences from Point Clouds via Multimodal Diffusion. CVPR 2024.
>
> [6] Khan et al., CAD-SIGNet: CAD Language Inference from Point Clouds using Layer-wise Sketch Instance Guided Attention. CVPR 2024.

---

### Official Review · Reviewer_jpmw · 2025-03-14

**Overall Recommendation:** 4

**Summary:**

This paper introduces CADFusion, a framework for Text-to-CAD generation that leverages LLMs and incorporates visual feedback to improve the quality and accuracy of generated CAD models. The core contribution is a two-stage training procedure that alternates between Sequential Learning (SL) and Visual Feedback (VF) stages. In the SL stage, an LLM (LLAMA-3-8b-Instruct) is fine-tuned on a dataset of paired textual descriptions and CAD parametric sequences (represented as text tokens), using a standard cross-entropy loss. This stage aims to teach the model to generate syntactically correct and logically coherent CAD commands. The VF stage then refines the model by incorporating visual information. Multiple CAD sequences are generated and rendered into visual representations (images). A VLM is employed to automatically score these rendered images based on multi-aspect criteria (shape quality, quantity, and distribution), creating preference data. DPO is then used to train the LLM to favor sequences that produce higher-scoring (visually preferable) CAD models, thus circumventing the non-differentiability of the rendering process. The authors propose an alternating training strategy, switching between SL and VF stages to preserve the benefits of both.

The paper presents both quantitative and qualitative results comparing CADFusion to two baselines: GPT-4o (with few-shot learning) and Text2CAD (Khan et al., 2024b). The authors utilize a variety of metrics, including F1 scores for sequence accuracy, Chamfer Distance, Coverage, MMD, JSD, and Invalidity Ratio for visual quality, and a VLM-based score to assess visual-textual correspondence. Human evaluations are also conducted. The main finding is that CADFusion outperforms both baselines on most metrics, particularly those related to visual quality. Ablation studies demonstrate the importance of the visual feedback stage and the alternating training strategy. The authors also contribute two datasets: one for sequential learning (text-sequence pairs) and another for visual feedback (preference pairs). The paper acknowledges limitations regarding the generation of very complex shapes.

In essence, the paper's main contributions include the combination of sequential and visual feedback training for Text-to-CAD, the use of DPO and LVM scoring to integrate visual information, and the demonstration of improved performance compared to existing methods.

## Update after Rebuttal
I thank the authors for their rebuttal. They have convincingly addressed several key concerns. They acknowledged the need for more precise claims and better highlighting of limitations regarding complex models, promising revisions. Their explanation of the VLM scoring process, including steps taken to ensure stability, was mostly clear. They also clarified the Text2CAD comparison, confirming they have results using identical prompts and will include these in the main paper. The provided clarifications and planned updates sufficiently address my main concerns, leading me to raise my score.

**Claims And Evidence:**

The paper puts forward several claims regarding the effectiveness of CADFusion, and for the most part, these are backed up by the experiments. For instance, the results in Table 1, along with the visual examples in Figure 5, make a good case that CADFusion generally outperforms the baselines, GPT-4o and Text2CAD, across a range of metrics. The ablation study in Table 2 also adds weight to this. Similarly, the benefits of the alternating training strategy are reasonably demonstrated by comparing different training variations within the ablation study.

However, there are a few places where the claims could be a bit more precise. The paper often uses phrases like "significantly improves performance," but without statistical significance tests, it's hard to know for sure if the improvements are truly significant or due to chance. Also, while the paper shows improvements, it's implicitly suggesting that CADFusion works well for all types of CAD models. This isn't entirely supported by the evidence. The examples are mostly simpler shapes, and the authors admit in the limitation section that the method struggles with more complex designs. This limitation should be brought forward more prominently, rather than implying broad applicability.

Another point concerns the VLM Score. Could different VLMs give different scores? Is the score sensitive to how the evaluation prompt is phrased? Finally, the comparisons with Text2CAD is not that solid, because they are using different prompts, as shown in Table 3.

In short, the core claims about the advantages of visual feedback and alternating training seem solid. But the claims about the extent of improvement and the general applicability need some toning down and more careful qualification, given the limited evaluation of complex models.

**Essential References Not Discussed:**

References seem adequate.

**Experimental Designs Or Analyses:**

They're mostly solid. The comparisons with GPT-4o and Text2CAD, using a mix of sequential and visual metrics, seem reasonable. The ablation studies in Table 2 also give us a good sense of how much each part (the visual feedback and the alternating training) actually contributes. The process of constructing datasets is clear.

**Methods And Evaluation Criteria:**

Overall, the proposed methods and evaluation criteria seem suited to the problem of Text-to-CAD generation. The core idea of combining sequential learning with visual feedback makes sense. CAD models have this dual nature – they need to be syntactically correct (valid sequences of commands) and visually accurate (matching the intended design). The two-stage approach, using an LLM for sequence generation and then refining it with visual feedback, directly addresses this. The use of DPO to handle the non-differentiable rendering process is a clever and appropriate technical solution. It's a good way to get around the problem of backpropagating through the rendering step. Similarly, using a VLM to automatically generate preference data seems a practical approach. Using F1-scores, Chamfer Distance, Coverage, MMD, JSD, and Invalidity Ratio for evaluation covers different aspects of visual quality and validity.

However, as mentioned before, an analysis of the VLM score's properties would strengthen the evaluation. Also, current metrics don't directly address aspects like manufacturability or design practicality. Furthermore, a more complete evaluation of the complex CAD models is needed.

In short, the methods and evaluation criteria make sense for the problem. The main areas for potential improvement are a more thorough analysis of the VLM score, and ideally, the inclusion of some evaluation related to design practicality and, most importantly, complex CAD models.

**Other Comments Or Suggestions:**

There are a few minor typos and grammatical issues, e.g,  Line 066: "texutal" -> "textual".

While the paper acknowledges limitations, a slightly more expansive discussion of future work would be valuable.

**Other Strengths And Weaknesses:**

The strengths and weaknesses of the paper have been covered in detail in the previous sections. Please refer to the "Claims and Evidence," "Methods and Evaluation Criteria," and "Experimental Designs and Analyses" sections for a discussion of these points.

**Questions For Authors:**

Please refer to the previous sections.

**Relation To Broader Scientific Literature:**

The paper does a reasonable job of positioning itself within the existing literature. The connections to broader work on LLMs, RLHF, and DPO are appropriately established.

**Theoretical Claims:**

This paper is primarily empirical and does not present theoretical claims or proofs.

---

> ### Author Rebuttal · Authors · 2025-03-31
>
> Dear reviewer,
>
> Thank you for your thoughtful comments and for recognizing our contributions, particularly in the design approach and experimental setup. We appreciate the opportunity to clarify the points you raised.
>
> ## Claim Could be a Bit More Precise
> We realize this problem and apologize for that. We will tone down relevant parts such as "significant performance improvements" in the updated version. The limitation about complex models (demonstrated in figure 11) will also be brought to the main body.
>
> ## Could Different VLMs Give Different Scores?
> First, VLM's scores depend on their base capabilities. We will add relevant discussions in the appendix. The followings are some discoveries we have made:
>
> 1. LLaVA (non One-Vison) cannot give correct scores. It tends to give very high scores (e.g., 9 or 10) or very low scores (e.g., 2 or 3) randomly.
> 2. How LLaVA-One-Vision (LLaVA-OV) scores rendered CAD objects is quite similar to how GPT-4o does.
>
> Second, the scores are partially sensitive to the formation of the evaluation prompts. Specifically, models randomly over-focus on aspects such as texture and colors of the rendered CAD objects, and score some rendered CAD objects lower than others based on that factors, which leads to inconsistency. Consequently, we instruct the VLMs not to focus on these aspects (details can be found in Line 752 and 800). After this modification, the scoring seems stable with prompts.
>
> ## Comparisons with Text2CAD
> We have the results where Text2CAD uses the same prompts as ours (i.e. the **solid** one). It is the comparison between row 3 and 4 in Table 3. The result we selected for presentation in Table 1 is the setup **favors Text2CAD the most**, which does not affect the fairness of our comparison. We will add the results with the same prompts to the main sections to make the comparison more comprehensive.

---

> > ### Comment · Reviewer_jpmw · 2025-04-05
> >
> > Thank you to the authors for their thorough rebuttal. After reviewing the rebuttal and considering the additional comments from other reviewers, I find the clarifications and updates compelling. As a result, I have raised my score accordingly.

---

### Official Review · Reviewer_jo2e · 2025-03-14

**Overall Recommendation:** 3

**Summary:**

This paper proposes a text-to-CAD model that leverages LLMs to generate CAD commend sequences as sequential signals. The authors use a pre-trained LLM as the backbone and perform SFT on CAD parametric sequences. They further use DPO to perform RL-based fine-tuning and introduce an LVM-based scoring pipeline to construct preference data.

**Claims And Evidence:**

1. The authors claim that they use DPO as the RL process to optimize the training.
2. An LVM is used to generate and rank the preference dataset.

The ablation study supports the claims.

**Essential References Not Discussed:**

HNC-CAD is able to perform partial CAD completion, which is similar to auto-regressive generation in this work and could be discussed.

[1] Xu, Xiang, et al. "Hierarchical neural coding for controllable cad model generation." arXiv preprint arXiv:2307.00149 (2023).

**Experimental Designs Or Analyses:**

The ablation studies support the claims of the modules.

A suggestion is that parital CAD completion could be a good application and experiment like in HNC-CAD

**Methods And Evaluation Criteria:**

The authors evaluated the method using F1 score, CD, COV, MMD, JSD, IR, LVM score and average rank. The former ones are common in the CAD generation tasks. LVM score and average rank are also reasonable as the authors use Large vision-language models to generate the preference dataset.

**Other Comments Or Suggestions:**

None

**Other Strengths And Weaknesses:**

Weaknesses
1. I still have concerns that the VLM-generated captions (even with human revision) could describe the CAD models well. Even if the training results are good, there may be some kind of overfitting to the dataset. It may still be hard for human users to generate such prompts that fit the VLM caption styles and also difficult for human users to correctly describe the CAD models purely in text prompts.

2. The captions are scale-invariant, so how do the users adjust the numerical parameters to scale different parts of the CAD models? Is it integrated to the LLMs are the users have to manually adjust them as a postprocessing step?

**Questions For Authors:**

1. Are the users able to modify the previously generated results with additional prompts?

2. What about the generation diversity regarding to the same text prompt?

**Relation To Broader Scientific Literature:**

This work incorporates powerful pre-trained LLMs into CAD generation. Could be a good start to the community.

**Theoretical Claims:**

Equation 1 is a commonly used cross entropy loss, and equation 2 is the loss from Direct Preference Optimization. No other theoretical claims are made in this paper.

---

> ### Author Rebuttal · Authors · 2025-03-31
>
> Dear reviewer,
>
> We sincerely appreciate your thoughtful comments and the opportunity to clarify the concerns raised.
>
> ## How Well Captions Describe CAD Models and Model Accomodates to Real Users
>
> During development, we considered similar concerns and would like to share our findings:
> * VLM-generated captions are effective for moderately detailed descriptions, such as structure, topology, and key component relationships. However, they struggle with precise numerical specifications, such as exact dimensions (e.g., "13.5 × 4.3 × 2"). Since designing is an iterative process, moderately detailed descriptions still provide substantial guidance to CAD modeling.
>
> * We provided guidelines to human users and empirically find that they quickly learned to interact with our model by following them. We provide more detail of the guideline and an example at the end of this response.
>
> ## Handling Numerical Parameters
> Currently, users should manually adjust numerical parameters as a post-processing step. In a pratical designing scenario, they normally create rough designs first and refine the numerical details iteratively. We believe our model provides a sufficient response automating the former step, and optimistically anticipate future work that supports the latter, aiming to fully automate the design process.
>
> ## More discussion for HNC-CAD
> We have discussed HNC-CAD in the introduction and related works as an unconditional CAD generation method, and cited it in our paper. Following your suggestion, we will add more discussion about HNC-CAD. Specifically,
> 1. HNC-CAD is also an autoregressive model. However, it trains a transformer **VQ-VAE** on **codebook** representations on **sequential** signals, while ours leverages a pretrained **LLM** (decoder only) on **stringified** representations on both **sequential and visual** signals.
> 2. The auto-completion task in HNC generates a random completion. It can be used jointly with the text-based CAD generation method. By providing a CAD sequence prefix and text guidence to the framework, we achieve a text-guided completion.
>
> ## User's Ability to Modify Generations with Additional Prompts
> We would put it as a promising future direction called text-based editing. It relies on different model capabilities from text-based generation. For instance, model may need to learn how to localize the place for edition based on additional prompts and make changes to that specific location. This requires not only text understanding but also CAD understanding.
>
> ## Generation Diversity Regarding the Same Prompt
> We did include exploration on this part! Please refer to Figure 10 in Page 18. It corresponds to Appendix C.7.
>
> [1] Khan et al., Text2cad: Generating sequential cad models from beginner-to-expert level text prompts, NeurIPS 2024.
>
> [2] Xu et al., SkexGen: Autoregressive Generation of CAD Construction Sequences with Disentangled Codebooks, ICML 2022.
>
> ## (Appendix) User Guidelines for Prompting
> A good prompt follows a structured description: (1) *shape overview*, (2) *shape details*, and (3) *shape applications*. Given the varying shape complexity, we encourage but do not enforce describing item (2) and (3). Below is an example caption retrieved from Figure 8, Row 2, Item 3, demonstrating this approach:
>
> ```
> [Shape Overview] The 3D shape consists of a large, flat rectangular slab with two evenly spaced, identical cylindrical protrusions extending vertically from its surface. [Shape Details (Optional)] The slab provides a stable base with significant length and width compared to its thin height, while the cylinders are relatively short and have small diameters. [Shape Applications (Optional)]The overall design is symmetrical and balanced, potentially serving as a mounting base or connector.
> ```
>
> We believe that such structured descriptions are clearly formatted and easy for users to interpret. If the reviewer finds it necessary, we are happy to include detailed instructions in the camera-ready version of the paper or the code repository upon release.

---

### Official Review · Reviewer_3nTM · 2025-03-19

**Overall Recommendation:** 2

**Summary:**

Authors build upon existing CAD data representation and introduce a novel visual feedback into text2cad. A dpo algorithm is used together with LVM-based scoring to improve the text2cad pipeline. Two new datasets are also proposed (text-cad pair dataset and preference dataset).

**Claims And Evidence:**

From table 2, the evaluation scores between SL (no visual feedback) and SL-VF(pro) is pretty close, I thought with the multi-aspect preference scores the valid ratio would greatly improve. But results do not seem to suggest so.

**Essential References Not Discussed:**

N/A

**Experimental Designs Or Analyses:**

Same as above, there is a lack of metric for benchmarking how well the text description controls the CAD generation.

**Methods And Evaluation Criteria:**

Evaluations for  whether or not the generation truly follows the text description is not obvious. LVM is not tuned to assess CAD model quality and the score might not be very reliable. The human ranking score is judging which output is the best out of four different models, but even the best model could still fail to adhere to some text constraints and this score will not reflect this.

**Other Comments Or Suggestions:**

N/A

**Other Strengths And Weaknesses:**

While I like the text2cad task, the technique contribution in this paper is limited. The CAD data representation is from previous work as well as the base dataset (DeepCAD) and the use of DPO. New contributions are limited to combining DPO with visual score feedback and  using VLM to annotate the DeepCAD models. But there are a lot of CAD structure and topology information not available from just a rendered image. So I am not sure if a VLM can capture all the information from image modality. The refined human annotation in figure 7 looks better, but going through the text annotations in figure 5, I find most of them are biased towards coarse shape of the CAD model (cylinder, rectangular, holes)…. This limits the text-cad pairs to very simple shapes (at least judging from the provided figures).

**Questions For Authors:**

1) 20k data is quite small for training a generative model. I wonder why authors did not annotate the full DeepCAD/SkexGen dataset. If human annotation is not scalable (as written in supplementary) then does it mean the VLM approach proposed in the paper can not reliablely generate high-quality annotations?

2) Why is there no cov/mmd/jsd scores for the other baseline models. Also why authors did not compare to DeepCAD/SkexGen/hnc-cad? Seems like data representation is the only difference here and it is not hard to retrain those models on the provided text annotations (all are open source).

3) I might have missed this but how exactly does VLM avoids collisions (figure 4)?

**Relation To Broader Scientific Literature:**

Text2CAD is generally an interesting topic in Computer-aided design. This has been less explored than other fields (image, video) due to lack of high-quality data. The paper is moving towards the right direction and demonstrates some early promising results to automate the design process.

**Theoretical Claims:**

N/A

---

> ### Author Rebuttal · Authors · 2025-03-31
>
> Dear reviewer,
>
> Thank you for your feedback. We appreciate the opportunity to clarify your concerns.
>
> ## SL(no VF) & SL-VF(pro)
> We are unsure what SL-VF(pro) refers to. We presume you are mentioning SF-VF(rpo). If so, we would like to point out that this is **not our main method**, but **an ablation** using existing techniques to regularize and stabilize DPO (Line 410). We included it to highlight the effectiveness of our *iterative visual-sequential training* (line 413-426).
>
> In Table 2, our main method is **SL-VFSL(5) in the last row**, where a considerable improvement on VLM score can be observed.
>
> ## Evaluation Criteria
> We regret to point this out, but there is a **factual misunderstanding**  in the statement *"Evaluations for whether or not the generation truly follows the text description is not obvious."*
>
> 1. In our human evaluation, the textual description are shown alongside the generation results, and annotators were asked to evaluate the alignment between the CAD model and text (Line 808). Thus, the human rankings directly reflect instruction following.
>
> 2. Second, our VLM score also supports this evaluation by design (Line 317). Reviewer `X4me` calls this "a reasonable score", which supports the soundness of our metrics.
>
> We would appreciate it if you could reconsider this claim after reviewing the referenced lines.
>
> ## Technical Contribution
> We would like to highlight that the motivation behind introducing DPO is itself a contribution. Observing **limitations of sequence learning**, we proposed using visual signals to adjust model preference: an approach has never been previously explored. Moreover, as the rendering process is not differentiable, directly encoding visual signals and backpropagating them is infeasible. Consequently, we propose DPO as a solution.
>
> Furthermore, integrating DPO into our setup is **not a trivial application** of existing techniques. Table 2 (Rows 3, 4) shows that directly applying DPO or standard stabilizing algorithms (RPO) does not yield strong results. We hypothesize that this is due to the **cross-modality** nature of our task. A significant portion of our work focused on overcoming challenges though SL-VFSL training.
>
> Additionally, we would like to highlight how other reviewers acknowledge our contributions. Reviewer `jpmw` calls our handling of the non-differentiable rendering process "clever and appropriate," and Reviewer `X4me` refers to our approach as "interesting" and a "nice way" of tackling the problem.
>
> ## Simple Shapes in Figures
> Figure 8, 9 includes CAD generations with more complex shapes and textual descriptions specifying CAD structure and topology details. We hope these results satisfy you.
>
> The reason Figure 5 contains 'simple' shapes is that we had to choose samples from the intersection of valid generations across models. We are happy to move some instances in Figure 7, 8 to the main manuscript at your wish.
>
> ## Data Size and Captioning
> 1. While we acknowledge that 20k human-annotated samples could be further expanded, we choose this setting for some reasons:
> - First, we annotated data and trained models iteratively. At 20k samples, our method already outperformed baselines (Table 1, main experiment).
> - Second, the cost of annotating 20k samples is affordable for research groups of any scale. By keeping the dataset size reasonable, we aim to ensure the reproducibility of our results and provide meaningful insights to the broader community.
>
> 2. VLM itself is reliable. The CADFusion variant trained on only VLM data with sequential signals (Table 4, row 2) outperforms baselines (Table 1, row 1, 2), demonstrating improvement.
>
> ## COV/MMD/JSD Scores
> The Text2CAD paper does not report these metrics. We attempted to compute them using our evaluation code but encountered challenges when applying it to Text2CAD's code and results. However, we have reported other metrics that effectively reflect the our method's efficacy.
>
> ## Comparing to DeepCAD/SkexGen/HNC-CAD
> We did not compare against them as none take text as input or claim text-based capabilities. Adapting their frameworks for textual input requires modifications that constitute a new research project rather than a baseline.
>
> Although this requirement is doubtful, we implemented a text-guided SkexGen variant by integrating a text encoder with SkexGen’s decoder for Text-to-CAD generation. Training on the same setups, our results are:
>
> | Method    |  COV  | MMD  |  JSD  |  IR   |
> |-----------|-------|------|-------|-------|
> | SkexGen   | 72.39 | 3.60 | 11.56 | 20.90 |
> | CADFusion | 90.40 | 3.49 | 17.11 | 6.20  |
>
> CADFusion outperforms SkexGen in multiple metrics, reinforcing its effectiveness.
>
> ## Collisions
> The term "collisions" refers to part intersections within a CAD model. VLM does not *explicitly* prevent collisions but evaluates their presence. This scoring mechanism benefits the subsequent visual-feedback stage, where we empirically observed an improvement in collision avoidance.

---

### Decision · Program_Chairs · 2025-05-01

**Decision:**

Accept (poster)

**Comment:**

This submission received one Accept, two Weak Accepts, and one Weak Reject. Three reviewers were positive overall, highlighting the novelty of combining sequential learning with visual feedback using DPO, the introduction of VLM-based scoring, and strong empirical results. The one negative review raised concerns about the dataset and annotation and baseline comparisons. The authors provided a detailed rebuttal, which the area chair found convincing. Reviewers are overall positive and the decision is an accept.